# Identification of Volatile Markers during Early *Zygosaccharomyces rouxii* Contamination in Mature and Immature Jujube Honey

**DOI:** 10.3390/foods12142730

**Published:** 2023-07-18

**Authors:** Yin Wang, Yuanyuan Huang, Ni Cheng, Haoan Zhao, Ying Zhang, Cailing Liu, Liangliang He, Tianchen Ma, Yankang Li, Wei Cao

**Affiliations:** Department of Food Science, College of Food Science and Technology, Northwest University, Xi’an 710069, China; wangyin2017@nwu.edu.cn (Y.W.); 15093665339@163.com (Y.H.); chengni@nwu.edu.cn (N.C.); zhaohaoan@nwu.edu.cn (H.Z.); naiyang2@163.com (Y.Z.); liucl@stumail.nwu.edu.cn (C.L.); heliangliang@nwu.edu.cn (L.H.); matianchen12311@163.com (T.M.); 20200085@nwu.edu.cn (Y.L.)

**Keywords:** jujube honey, HS-SPME-GC-MS, *Zygosaccharomyces rouxii*, VOC

## Abstract

Osmotolerant yeasts are considered one of the major contaminants responsible for spoilage in honey. To address the signature volatile components of jujube honey contaminated by *Zygosaccharomyces rouxii*, headspace solid-phase microextraction-gas chromatography-mass spectrometry (HS-SPME-GC-MS) and chemometrics analyses were used to analyze the variation of volatile substances during early contamination of mature and immature jujube honey. Undecanal, methyl butyrate, methyl 2-nonenoate, methyl hexanoate, and 2-methyl-3-pentanone were identified as signature volatiles of jujube honey contaminated with *Z. rouxii*. In addition, methyl heptanoate, 2,6,10-trimethyltetradecane, and heptanal were identified as potential volatile signatures for immature jujube honey. The R^2^ and Q^2^ of OPLS-DA analyses ranged from 0.736 to 0.955, and 0.991 to 0.997, which indicates that the constructed model was stable and predictive. This study has demonstrated that HS-SPME-GC-MS could be used to distinguish *Z. rouxii*-contaminated jujube honey from uncontaminated honey based on variation in VOCs, and could provide theoretical support for the use of HS-SPME-GC-MS for the rapid detection of honey decomposition caused by *Z. rouxii*, which could improve nutritional quality and reduce economic losses.

## 1. Introduction

Jujube honey is a bulk honey native to China that exhibits an amber color, unique aroma, and a neutral pH [1]. In addition to high concentrations of phenolic substances, hydrogen peroxide, and proteins, it possesses potent antioxidant properties and medicinal value, rendering it unsuitable for the survival of most microorganisms, except for osmotolerant yeasts [2,3]. Bees completely ferment jujube honey until it reaches maturity [4]. During honey’s maturation, its antibacterial activity and antioxidant stability increase substantially, its water content decreases, and its quality and nutritional value increase significantly [5]. Currently, some beekeepers purchase immature honey and sell it as mature honey in order to reduce the ripening time. However, immature jujube honey contains approximately 25% water and is highly susceptible to contamination by osmotolerant yeast, which significantly diminishes the product’s quality and nutrient content. This practice has been identified as honey counterfeiting on markets, which has a significant impact on the international circulation and the industrial development of jujube honey [1,6].

Osmotolerant yeast contamination is difficult to detect in the early stage, and in the middle and late stages, it can significantly alter honey’s taste and nutritional value, which may result in sourness and even serious spoilage. This contamination can not only result in economic losses for businesses, but can also negatively impact international trade [7,8,9]. The CAC standard requires that honey must not ferment or undergo flavor changes during storage, sale, or transport. Therefore, osmotolerant yeasts are listed as one of the major spoilage bacteria to be controlled, and have been included as an important test indicator for import and export inspection of honey by various countries over the past decade [10].

At present, the most widespread method for detecting osmotolerant yeast in honey is plate counting, which is time consuming and susceptible to initial false-negative results due to operational issues [11]. High-performance liquid chromatography with mass spectrometry (HPLC-MS) is a highly sensitive technique, but its detection targets are typically secondary metabolites of yeast fermentation, which frequently cause severe product degradation [12]. Volatile organic compounds in food are substances with a high vapor pressure at room temperature and standard atmospheric pressure, which are emitted from food surfaces and have a significant effect on a number of important food quality characteristics [13]. About 600 volatile organic compounds (VOCs) have been identified in honey, most of which are derived from nectar plant sources as well as honey pollinators during honey production via enzymatic catalysis and synthetic transformation [14,15]. Therefore, the volatile components of the same type of honey are relatively stable [16]. Alcohol, acid, aldehydes, alkanes, ketones, terpenes, phenols, esters, and other volatile flavor substances are abundant in jujube honey. Hexyl alcohol, heptyl acetate, heptyl aldehyde, methyl heptylate, cedarene, linalsol, and terpenol are useful indicators for distinguishing jujube honey from other sources of single-flower honey [15]. Due to the high sensitivity of VOC detection, the discrimination of microbial contamination based on VOCs has become a reliable tool for evaluating the quality and safety characteristics of food products, and specific VOCs can also be used as biomarkers to provide crucial information for food traceability [17,18].

Currently, gas chromatography-ion mobility spectrometry (GC-IMS), and electronic nose gas chromatography-sniffing-mass spectrometry (GC-O-MS) are utilized to identify volatile substances in honey. However, these techniques typically suffer from low stability, high environmental requirements, and susceptibility to the effect of the honey’s sugar matrix [19]. Headspace solid-phase microextraction-gas chromatography-mass spectrometry (HS-SPME-GC-MS) can overcome the shortcomings of these methods, as HS-SPME not only has high extraction efficiency and does not require complex pretreatment, but can also be coupled with gas-phase and mass spectrometry-type analytical instruments [20,21]. And microbial detection methods based on VOCs are generally accepted for fermented food flavor analysis, disease diagnosis, rotten food detection, and even antimicrobial resistance bacteria detection [21,22]. When combined with the headspace phase, it can safeguard the extraction head from the honey’s sugar matrix. GC-MS also possesses the high separation capacity and high detection efficiency of gas chromatography, accurate quantification and characterization, mass spectrometry identification, and a computer that can process a large volume of data [23,24]. In recent years, HS-SPME-GC-MS has demonstrated tremendous promise in the field of honey safety and rapid detection.

VOC-based assays in honey are mainly focused on the identification of floral sources, ground sources, and aroma changes during different stages of flowering and maturity detection [1]. At distinct contamination stages, the detection and characterization of osmotolerant yeast contamination in honey based on volatile organic compounds (VOCs) has not been reported. Additionally, it is still not yet known whether there are differences in the volatiles produced by osmotolerant yeast contamination between mature and immature honey.

In this research, *Zygosaccharomyces rouxii*, which was isolated from honey in our previous studies, was used to artificially contaminate immature and mature jujube honey. To address the signature volatile components of jujube honey contaminated by *Zygosaccharomyces rouxii*, HS-SPME-GC-MS was utilized to examine the various volatile substances present during early contamination of mature and immature jujube honey. This study could provide theoretical support for the use of HS-SPME-GC-MS for the rapid detection of honey spoilage and deterioration, which has the potential to enhance nutritional quality and reduce economic losses.

## 2. Materials and Methods

### 2.1. Strain and Chemicals

*Zygosaccharomyces rouxii* XD4-1, which was isolated from honey samples, was stored at −80 °C in our lab. Yeast extract, peptone and agar were purchased from Beijing Land Bridge Technology Co. (Beijing, China), and NaCl, glucose was purchased from Yuanye Biological Technology Co., Ltd. (Shanghai, China). Benzophenone (purity ≥ 99%) was purchased from Rhawn (Shanghai, China). N-alkanes series (C6–C40) (purity > 98%) were obtained from Sigma-Aldrich (St. Louis, MO, USA).

### 2.2. Honey Sample Collection

Mature jujube honey (MH) was sampled from Laofengnong Co. (Xi’an, China). The immature jujube honey (IH) with a storage period of 3 days was sampled from cooperative beekeepers from Jiaxian county, Yulin City, Shaanxi province. The sampling method for immature jujube honey was performed according to Sun [16]. To ensure that the honey samples used in this study were not contaminated with other yeasts, the plate-counting method with YPD was used to test for the presence of *Z.* spp. in both mature and immature jujube honey samples. Finally, all of the fresh honey samples were stored at −20 °C before use. DVB/CAR/PDMS solid phase microextraction column and DB-5 chromatographic separation column were purchased from Agilent Technologies, Santa Clara, CA, USA.

### 2.3. Sample Preparation

*Z. rouxii* XD4-1 was cultured on yeast extract peptone dextrose agar (YPD, 1% yeast extract, 2% peptone, 4% glucose, 2% agar) at 28 °C for 2–4 days. A single colony was streaked and suspended in 1 mL sterile physiological saline. The *Z. rouxii* cells were collected via centrifugation at 5000 rpm for 5 min, and then washed twice with sterilized distilled water. Cell suspension was diluted to 10^−3^ and 10^−4^ concentrations via the gradient dilution method. The final concentration of the *Z. rouxii* solution was adjusted to around 1000 CFU/mL and 100 CFU/mL. YPD agars were plated with two hundred milliliters of culture dilutions for colony counting, and three times for each concentration in order to eliminate random error. For high-concentration groups (H), a total of 100 μL 10^−3^ *Z. rouxii* suspension was added to 12 samples of 20.0 g MH and 12 samples of 20.0 g IH, and then incubated at 25 °C. At 3, 7, 15, and 30 days; one group (3 samples) was taken out and stored at −80°C before further testing. The same method was performed to produce low-concentration groups (L) of jujube honey. After 30 days of incubation, mature and immature jujube honey without yeast inoculation were used as blank control (CK) in order to determine the volatile substances.

### 2.4. HS-SPME-GC-MS Analysis

The method was modified slightly based on the work of Zhu et al. [15]. Specifically, by adding 0.2 g of NaCl and 2 mL of distilled water to 2.0 g of honey sample, and then dissolving it in a 20-mL vial with PTEE silica gel spacer. Extraction column type DVB/CAR/PDMS was selected in the headspace solid-phase microextraction (HS-SPME) method. The extraction conditions were as follows: equilibrium temperature of 45 °C, and a time of 15 min; sample puncture depth of 30 mm, and a speed of 20 mm/s; extraction temperature of 45 °C, and a time of 30 min; stirring speed of 600 rpm; and desorption temperature of 250 °C, and with a time of 10min. The extraction column aging temperature and time were 180 °C and 5 min, respectively. Regarding mass spectrometry conditions, DB-FFAP (60 m × 250 μm × 0.25 μm, Agilent) was selected as the separation column, and equipped with Agilent 7890B GC and Agilent 5977B GC/MS, with a helium carrier gas flow rate of 1.8 mL/min and no splitting. The GC-heating procedure was as follows: the initial temperature was 45 °C for 1 min, then increased to 225 °C at 3 °C/min and held for 3 min, and finally increased to 240 °C at 20 °C/min and held for 5 min. The inlet temperature was 250 °C and the column chamber temperature was 45 °C. The mass spectrometry conditions were as follows: at 70 eV, the *m*/*z* scanning range was 50~350, the ion source temperature was 240 °C, and the quadrupole temperature was 150 °C. All samples were performed in triplicate.

The C6–C40 n-alkane standard was selected as the carbon standard, and analyzed via GC-MS, according to the sample determination procedure. The LRI index of the target compound was calculated based on the retention time of the peak of the separated compound, and the qualitative analysis of the compound was performed by querying the RI index of the NIST 17 database. The LRI index was calculated as follows:$$\mathrm{LRI}=100\times \left[\frac{\log_{10}^{x_{i}}-\log_{10}^{x_{n}}}{\log_{10}^{x_{n+1}}-\log_{10}^{x_{n}}}+n\right]$$

The *x_i_* was the peak retention time of the target compound; *x_n_* and *x_n+1_* were the retention times corresponding to the two adjacent n-alkane standards before and after the target compound.

The benzophenone added to the sample was used as an internal reference, and the ratio of the peak area of each compound in the sample to the peak area of the internal standard benzophenone was calculated for the semi-quantitative analysis of the target compounds.

### 2.5. Data Analysis

In this study, all results were expressed as mean ± standard deviation (SD) (*n* = 3). One-way significant differences (*p* < 0.05) and correlation coefficients were calculated using SPSS 25.0 software (SPSS Inc., Chicago, IL, USA) [25]. Principal component analysis (PCA) and orthogonal partial least squares discriminant analysis (OPLS-DA) were performed using SIMCA 14.1 (MSK Umetrics AB, Umeå, Sweden) to screen for differential compounds using VIP (Variable Important on the Projection) values and *p*-values obtained from OPLS-DA. Cluster analysis of volatile organic compounds was performed using the MetaboAnalyst 5.0 website (https://www.metaboanalyst.ca/ accessed on 23 June 2023), and all data were transformed via log transformation (base 10) and normalized to create a heat map. Venn diagrams were produced through the Evenn website (http://www.ehbio.com/test/venn/#/ accessed on 25 June 2023) and Adobe Illustrator 2023 software (version 27.7, Adobe Inc., San Jose, CA, USA).

## 3. Results and Discussion

### 3.1. Volatiles in Zygosaccharomyces rouxii-Contaminated Jujube Honey Identified via HS-SPME-GC–MS

A total of 22 and 25 compounds were detected in MH and IH before and after *Z. rouxii* contamination, respectively, with slightly more VOCs in IH than MH. Alcohols (MH:4, IH:4), aldehydes (MH:4, IH:5), ketones (MH:1, IH:1), esters (MH:7, IH:8), aromatic hydrocarbons (MH:3, IH:3), alkanes (MH:2, IH:3), and furans (MH:1, IH:1) constituted the majority of the volatile compounds (Table 1 and Table 2). The most abundant volatile organic compounds in jujube honey were esters, followed by alcohols and ketones. The categories of volatile substances in MH and IH were comparable, with the exception of heptanal, methyl heptanoate, and 2,6,10-trimethyltetradecane, which were detected only in IH. Other volatile organic compounds were detected in both MH and IH. Both MH and IH showed little variation in VOC species at different concentrations of *Z. rouxii* contamination. (*E*)-2-Octenal was the specific VOC, which was only detected in low-*Z. rouxii*-concentration groups (L) in both MH and IH. α-Methyl-α-[4-methyl-3-pentenyl]oxiranemethanol and methyl 13-octadecenoate were specific VOCs that were only detected in high-Z. rouxii-concentration groups (H) in both MH and IH. Toluene, octane, *p*-xylene, *o*-xylene, nonane, nonanal, decanal, methyl nonanoate, and methyl palmitate were stably present in MH and IH, and these substance types were not adversely affected by the presence of *Z. rouxii* (Table 1 and Table 2).

### 3.2. Identification of Characteristic VOCs by HS-GC-IMS

#### 3.2.1. Alcohols

Alcohols, which have a floral and fruity fragrance, are one of the most common aroma components in honey and are produced primarily via the amino acid lipoxygenase pathway [26]. Four alcohols were detected in all four groups (HMH, HIH, LMH, and LIH): cis-5-ethenyltetrahydro-α,α-5-trimethyl-2-furanmethanol, linalool, cedrol and α-methyl-α-[4-methyl-3-pentenyl] oxiranemethanol. Among them, cis-5-ethenyltetrahydro-α,α-5-trimethyl-2-furanmethanol was stable in both mature and immature jujube honey, but its substance content fluctuated over time, possibly due to substance transformation (Figure 1, Table 1, Appendix A). α-Methyl-α-[4-methyl-3-pentenyl]oxiranemethanol was stable; its relative levels did not change substantially over time, and it was unaffected by *Z. rouxii* (Figure 1, Table 1), but it was not detected in honey contaminated with low concentrations of *Z. rouxii* (L groups).

The relative concentration of linalool progressively increased in the presence of *Z. rouxii*, and detection occurred 3 and 30 days after contamination in the H and L groups, respectively, which likely corresponded to the initial dose of inoculum (Figure 1, Table 1, Appendix A). Linalool was not detected in the CK of MH, but it was detected in small amounts in the IH blank samples. It was observed that the relative concentration of linalool increased and then decreased over time, and the transmutation of the substance appeared to be associated with this variation. Linalool possessed a floral, fragrant, and grape-like flavor profile [27], is a common byproduct of yeast fermentation, and is a precursor to numerous alkenoids. Linalool was readily transformed by yeast-related enzymes and in acidic environments [28], and its relative concentration was substantially and positively correlated with the substrate’s available nitrogen source [27]. Consequently, the relative concentration of linalool and its oxides tended to vacillate during the initial phase of microbial fermentation in immature jujube honey with a high water content and an unstable environment. According to Gaglio [29], the relative content of linalool and oxidized linalool in honey under *Z. rouxii* contamination fluctuates over time for the first 9 days, but shows an overall progressive increase after 13 days, which is consistent with the results of the present study [29]. Cedrol was detected in both L and H groups of mature jujube honey 7 days after contamination (Figure 1, Table 1). Its relative levels increased over time in the HMH group, and increased and then decreased in the LIH group. Although cedrol was detected in the immature jujube honey blanks, its relative content was consistent with that of mature jujube honey.

#### 3.2.2. Esters

Esters have a distinct fruity and floral aroma, and they are formed in honey primarily through the oxidative degradation of unsaturated fatty acids, fatty acids, and straight- or branched-chain carboxylic acids derived from the amino acid pathway. They can also be produced by the enzymatic action of alcohols produced by yeast, which contribute significantly to the formulation of particular aromas [30,31]. Seven esters were detected in both mature and immature jujube honey, with the exception of heptanoic acid methyl ester, which was detected only in immature jujube honey (Figure 1, Table 1). Methyl ester of heptanoic acid was only detectable in IH samples. Its relative concentration decreased progressively with the duration of *Z. rouxii* contamination and vanished 7 days after contamination in both HIH and LIH groups. Methyl nonanoate, methyl 11-methyldodecanoate, and methyl palmitate were found in both mature and immature jujube honey CK samples. There was no discernible pattern between the three groups regarding the relative proportions of these substances (Figure 1, Table 1, Appendix A). Methyl butyrate, methyl 2-nonenoate, and methyl 13-octadecenoate were volatile compounds absent in both mature and immature jujube honey CKs, and they accumulated gradually over time with *Z. rouxii* contamination (Figure 1). Methyl butyrate has an apple-like, fruity aroma and is the characteristic volatile of strawberry and other fruits after maturation and storage [32,33]. The vast majority of yeasts are capable of producing butanoic ester compounds, of which butanoic acid is primarily derived from the degradation of lactose and liberated amino acids or the oxidation of ketones, esters. Yeasts produce compounds such as methyl butyrate and ethyl butyrate using related enzymes [34]. In our study, 7 and 30 days after *Z. rouxii* contamination, methyl butyrate was detected in both HMH and IMH. And for immature groups, it was detectable at 15 days in the HIH and 30 days in the LIH. Butanoic acid ethyl ester was detected in honey that had been stored for an extended period of time, but methyl butyrate was not [35]. During yeast fermentation, the concentration of methyl esters of fatty acids was influenced by fermentation temperature and time [18,36], which may account for the disparate results observed in our study. The methyl 2-nonenoate was not detected in the CK of any of the four sample groups, but it was detected three days after contamination. Its relative concentration increased as the duration of *Z. rouxii* contamination increased. One can infer that *Z. rouxii* converted the compound from methyl 2-nonenoate. In contrast, nonanoic acid is the most abundant medium-chain FFA in royal jelly, with a small amount of 2-nonanoic acid also present as regio-isomers of nonanoic acid [32]. Therefore, the substance may result from the esterification reaction initiated by *Z. rouxii*. Nonetheless, additional verification was still required. Methyl 13-octadecenoate was only detected 15 days after both MH and IH were contaminated with high concentrations of *Z. rouxii*. The synthesis of methyl 13-octadecenoate has been shown to contribute to the survival of eukaryotic cells under hyperosmotic conditions [37], although there was no literature directly demonstrating a link between *Z. rouxii*’s stress resistance and methyl 13-octadecenoate. In hyperosmotic conditions, however, *Z. rouxii*’s fatty acids were predominantly C18 and C16. There was a significant positive correlation between oleic acid (C18:1) content and osmolarity [38]. Therefore, it was hypothesized that the compound could be formed via the esterification of oleic acid derived from *Z. rouxii* and honey alcohol after fermentation. However, detailed reasons for this will require further investigation.

#### 3.2.3. Aldehydes and Ketones

Aldehydes and ketones are typical aromatic components of honey with a fruity aroma. Among the five aldehydes, nonanal and decanal were stably detected in both mature and immature jujube honey (Figure 1, Table 1). Nonanal and decanal have typical rose and citrus scents, giving the honey a distinct floral and fruity aroma. The relative content of nonanal reached the highest in the four groups about 7 days after *Z. rouxii* contamination, and then decreased. The relative concentration of decanal increased progressively in mature honey, whereas it decreased and then increased in immature honey. It was conceivable that immature jujube honey contained more water and displayed more active material transformation. Zhu et al. [15] found that nonanal and decanal accounted for a higher proportion of jujube honey VOCs as its signature volatile components [15]. A similar result could also be found in the present study, and although there were some differences between the VOCs of mature and immature jujube honey, nonanal and decanal were still their signature volatile components. Undecanal has a fatty wax aroma, and some studies have found that undecanal has been detected in a variety of honeys including jujube honey and lavender honey [15]. However, in the present study, it was not detected in both mature and immature jujube honey before *Z. rouxii* contamination. Moreover, with the increasing number of days after contamination, the relative content of undecanal in all four groups initially increased and then decreased, among which in the HMH and the HIH groups could be detected from 3 days. (*E)*-2-Octenal, which is considered an undesirable flavor in food because of its cardboard-like taste [39], was not present in both mature and immature jujube honey CK samples. In L groups, it was detected within 3 days after contamination in both LMH and LIH. However, no (*E*)-2-octenal could be detected in H groups (HMH, HIH). Yeast fermentation could significantly reduce the production of (*E*)-2-octenal, which could partially explain why this substance was not detectable at later stages of contamination in high-*Z. rouxii*-concentration groups [39]. (*E*)-2-Octenal is an aliphatic aldehyde, which originates primarily from the oxidation of linoleic and oleic acids or microbial degradation [40,41]. The aroma is described as green, cucumber-like, and fatty [5] and has commonly been found in grape varieties [42], oolong tea [43], and other foods. However, it tends to impart a unpleasant flavor similar to a fatty flavor, and it also acts as a signature volatile in foods in long-term storage [40]. 2-Methyl-3-pentanone was the only ketone detected in this study and was rarely detected as a representative volatile substance in foods. It was not detected in any of the four groups of CK before *Z. rouxii* inoculation, and its relative content accumulated gradually with the prolongation of the contamination. Specifically, the substance was detected in H groups after 3 days of contamination, while it needed 30 days in L groups. 2-Methyl-3-pentanone may be a unique product formed by the degradation of oils, alkanes in foods or the sterol fraction of phytosterols by the action of hydrogen peroxide [44]. Except for the chain alkanes in honey, jujube honey was not found to be rich in sterols. However, some sterols in jujube honey might be brought in from royal jelly by honeybee activity, and high concentrations of sterols were important precursors for honeybee molting hormones and cellular membranes [45]. On the other hand, the enrichment of hydrogen peroxide in jujube honey might be an important influencing factor in the formation of 2-methyl-3-pentanone. However, we did not detect this substance in all CK samples stored for 30 days under the same storage conditions. At the same time, its relative content and contamination time were significantly and positively correlated with the initial inoculum, so it was presumed that the production of this substance was related to the contamination of yeast. However, the specific mechanism that triggered the change still needs to be further explored.

#### 3.2.4. Hydrocarbons and Furan

The majority of the hydrocarbons in honey are derived from animal-derived beeswax [46]. In this study, five hydrocarbons (toluene, *p*-xylene, *o*-xylene, octane and nonane) and one furan (2-methyl-5-pentyltetrahydrofuran) were detected in CKs and all samples during 30 days of contamination (Figure 1, Table 1). Trimethyl-2,6,10-tetradecane was detectable in immature jujube honey CKs. Nonetheless, as the contamination persisted, its relative concentration decreased steadily, becoming undetectable after 3 to 7 days. Since this substance was not detected in ripe jujube honey, its relative decrease could not be attributed to the yeast contamination. It was likely related to the transformation of substances during honey ripening, but this must be confirmed through additional experiments. 2-Methyl-5-pentyl-tetrahydrofuran has a caramel flavor, and its relative concentration in the four groups increased with time after contamination. The variation may be a result of *Z. rouxii*’s ability to degrade free amino acids via the ehrlich pathway and liberate higher alcohols and 4-hydroxyfuranones [38].

### 3.3. Determination of Violate Markers in Z. rouxii-Contaminated Jujube Honey

For *Z. rouxii*-contaminated mature and immature jujube honey, the changes of volatiles and their signature markers were further obtained based on chemometrics analysis performed via SIMCA14.1 software. Samples at a different contamination periods could be clearly separated based on the VOC variation of *Z. rouxii*-contaminated mature and immature jujube honey, via PCA analysis. There was no overlap between the samples from different time periods after contamination based on the first two principal components (Figure 2). To further analyze the differential volatiles before and after *Z. rouxii*-contamination in jujube honey, supervised multivariate orthogonal partial least squares-discriminant (OPLS-DA) analysis was implemented among the HMH, HIH, LMH, and LIH groups. The results showed that OPLS-DA identified as clear a distinction between different groups of samples as PCA, which indicated that the model was significantly effective in classifying samples at this inoculum level, with significant differences between groups (Figure 3). Specifically, in this study R^2^_(HMH)_ = 0.845, Q^2^_(HMH)_ = 0.992; R^2^_(LMH)_ = 0.955, Q^2^_(LMH)_ = 0.987; R^2^_(HIH)_ = 0.736, Q^2^_(HIH)_ = 0.997; R^2^_(LIH)_ = 0.892, Q^2^_(LIH)_ = 0.991, indicated that the constructed model was stable and predictive. As a result, it could be used to distinguish the *Z. rouxii*-contaminated jujube honey among different contamination times. According to the OPLS-DA biplots for mature jujube honey (Figure 3a,b), undecanal and methyl 2-nonenoate mainly contributed to the differentiation of samples 7 days after contamination. Methyl 11-methyldodecanoate mainly contributed to the differentiation around 15 days. Methyl butyrate and methyl hexanoate were the characteristic volatiles around 30 days. And 2-methyl-3-pentanone was the characteristic volatile around 15 days and 30 days in the HMH and LMH groups, respectively.

For immature jujube honey, methyl heptanoate mainly contributed to the differentiation of CK. Methyl hexanoate and methyl 2-nonenoate mainly contributed to the differentiation of 7 days in the HIH group and 15 days in the LIH group. Methyl butyrate was the main contributor to the 15-day discrimination in HIH and the 30-day discrimination in LIH. Methyl hexanoate was the main contributor to the 30-day discrimination in HMH and 15-day discrimination in LIH, respectively (Figure 3c,d). In any random permutation at the left end, moreover, all values were substantially lower than the original values at the right end (Appendix A). The slopes of R^2^ were greater than 0, and the intercepts of Q^2^ were less than 0 (HMH: R^2^ = 0.109, Q^2^ = −0.821; LMH: R^2^ = 0.146, Q^2^ = −0.808; HIH: R^2^ = 0.101, Q^2^ = −0.838; LIH: R^2^ = 0.100, Q^2^ = −0.804), indicating that the model exhibited a good fitness and acceptable predictability.

The variable importance in the projection (VIP) quantified the contribution of each component to the classification. Among them, VOCs with VIP values > 1 could be identified as potential characteristic markers. A total of 14 volatile components a with VIP > 1 were detected in the four groups (Figure 4). Undecanal, methyl butyrate, and methyl 2-nonenoate were present with a VIP > 1 in all four groups except CKs. Their relative contents varied in a more consistent pattern after contamination. Therefore, the three substances could be considered to be the signature volatiles of *Z. rouxii* contaminated with jujube honey (Figure 5). While methyl hexanoate, and 2-methyl-3-pentanone exhibited VIP>1 in only three groups, their relative contents varied in a more uniform pattern, and they may also be considered signature volatiles when used in combination with OPLS-DA analysis (Figure 5). Methyl heptanoate, 2,6,10-trimethyltetradecane, and heptanal were VOCs specific to immature jujube honey, with VIP > 1 and a uniform variation pattern during contamination. While they were not detected in MH, the above three compounds could be classified as potential signature volatiles of *Z. rouxii*-contaminated IH.

### 3.4. VOC Variation Analysis among Different Z. rouxii Concentration Contamination in MH and IH

In general, a total of 20 volatile compounds (VOCs) were found to be common in all four groups before and after contamination with *Z. rouxii* (Figure 6). Among them, there were slightly more VOCs in IH than in MH, and the VOCs in the *Z. rouxii* H groups were slightly higher than those in the L groups. Methyl heptanoate, 2,6,10-trimethyltetradecane, and heptanal were specific VOCs to IH. Methyl 13-octadecenoate and α-Methyl-α-[4-methyl-3-pentenyl]oxiranemethanol were VOCs specific to the high-concentration contaminated samples, and (*E*)-2-octanal, was the VOC specific to the low-concentration contaminated samples (Table 1, Figure 1). After contamination by *Z. rouxii*, MH VOC categories were relatively stable and the overall variations were more regular. Although IH possessed more compound types, its substance types and relative contents were more volatile and fluctuated with the prolongation of storage.

In this analysis, esters were the most diverse group of contaminants, and the majority of them were methyl esters. Certain compounds were not previously reported in honey, and their relative concentrations were positively correlated with the time of contamination and the initial level of contamination by *Z. rouxii*. It was conceivable that these compounds derive from the TCA cycle of yeast’s oxidative breakdown of carbohydrates and fatty acids in honey to the corresponding acids. In the presence of the relevant enzymes [38], it then produced by an esterification reaction with methanol that accompanied ethanol production during *Z. rouxii* fermentation. It is interesting to observe that yeast fermentation produced the majority of ethanol in the previous study. Most of the acid produced during fermentation was in the form of ethyl esters, which contradicts the findings of the present study. Martnez-Garca et al. [36] discovered that the fermentation temperature and fermentation duration influenced the amount of methyl esters of fatty acids in yeast-fermented sparkling wines [36]. This study’s fermentation simulated the temperature of yeast-contaminated honey in its natural state. Consequently, the fermentation temperature was lower than that of wine fermentation, and the fermentation period was shortened. Therefore, we hypothesize that the production of significant quantities of methyl esters may be related to storage temperature and time, but further investigation is required to determine the precise cause.

## 4. Conclusions

Based on HS-GC-MS analysis, there were no significant differences in VOC types between MH and IH. The VOCs and relative contents of jujube honey changed significantly before and after *Z. rouxii* contamination. There were slightly more VOCs in IH than MH, and the changes in VOCs and relative contents were positively correlated with the amount of initial *Z. rouxii* inoculation. Alcohols, aldehydes, ketones, esters, aromatic hydrocarbons, alkanes, and furans constituted the majority of the volatile compounds. Chemometrics analysis revealed that HS-SPME-GC-MS could effectively distinguish *Z. rouxii*-contaminated jujube honey from uncontaminated honey based on VOC changes, which were less affected by maturity and initial contamination amount and could be detected as early as 3 days after contamination. Five volatile organic compounds (undecanal, methyl butyrate, methyl 2-nonenoate, methyl hexanoate, and 2-methyl-3-pentanone) were identified as signature volatiles of jujube honey contaminated with *Z. rouxii*. Methyl heptanoate, 2,6,10-trimethyltetradecane, and heptanal were also identified as potential signature volatiles for immature jujube honey contaminated with *Z. rouxii*.

## Figures and Tables

**Figure 1 foods-12-02730-f001:**
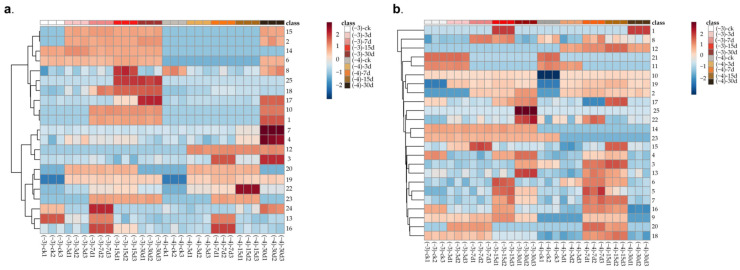
Variation of volatile components in mature and immature jujube honey during *Z. rouxii* contamination via HS-SPME-GC-MS. (**a**) Mature jujube honey; (**b**) immature jujube honey; “−3” for contamination with high-concentration *Z. rouxii*; “−4” for contamination with low-concentration *Z. rouxii*. (1) Methyl butyrate; (2) 2-Methyl-3-pentanone; (3) toluene; (4) octane; (5) *p*-Xylene; (6) 2-Methyl-5- pentyltetrahydrofuran; (7) *o*-Xylene; (8) nonane; (9) heptanal; (10) methyl hexanoate; (11) methyl heptanoate; (12) (E)-2-Octenal; (13) cis-5-Ethenyltetrahydro-α,α-5-trimethyl-2-furanmethanol; (14) α-Methyl-α-[4-methyl-3-pentenyl]oxiranemethanol; (15) linalool; (16) nonanal; (17) decanal; (18) methyl nonanoate; (19) methyl 2-nonenoate; (20) undecanal; (21) 2,6,10-Trimethyltetradecane; (22) methyl 11-methyldodecanoate; (23) cedrol; (24) methyl palmitate; (25) methyl 13-octadecenoate.

**Figure 2 foods-12-02730-f002:**
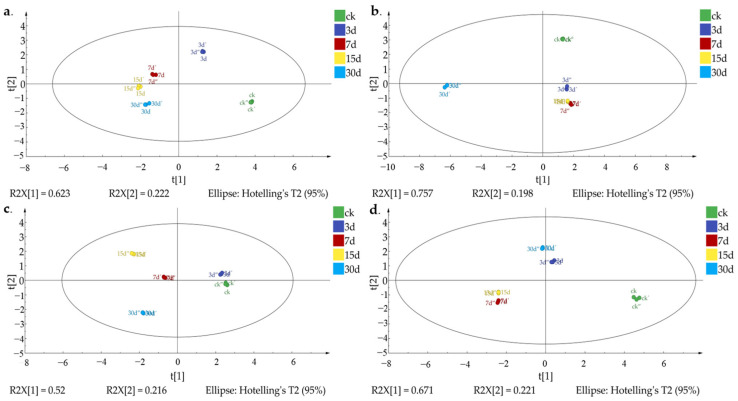
PCA score plot of mature and immature jujube honey in different contamination stages. (**a**) Mature jujube honey contaminated with high *Z. rouxii* concentration (HMH); (**b**) mature jujube honey contaminated with low *Z. rouxii* concentration (LMH); (**c**) immature jujube honey contaminated with high *Z. rouxii* concentration (HIH); (**d**) immature jujube honey contaminated with low *Z. rouxii* concentration (LIH).

**Figure 3 foods-12-02730-f003:**
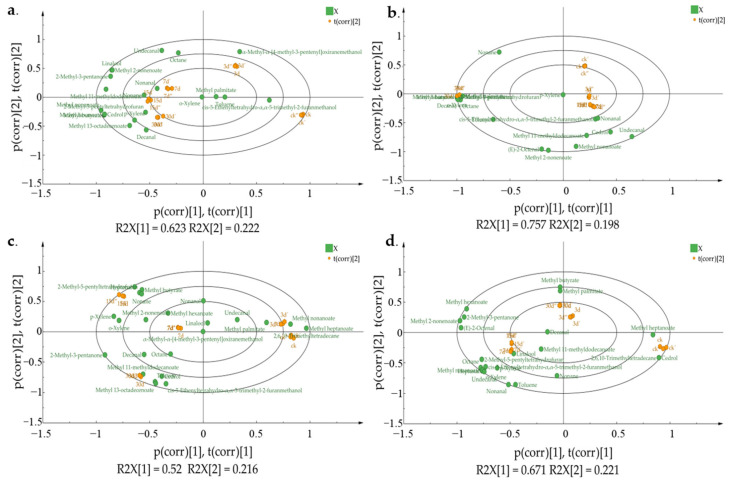
OPLS-DA biplot of mature and immature jujube honey in different contamination stages. (**a**) HMH; (**b**) LMH; (**c**) HIH; (**d**) LIH.

**Figure 4 foods-12-02730-f004:**
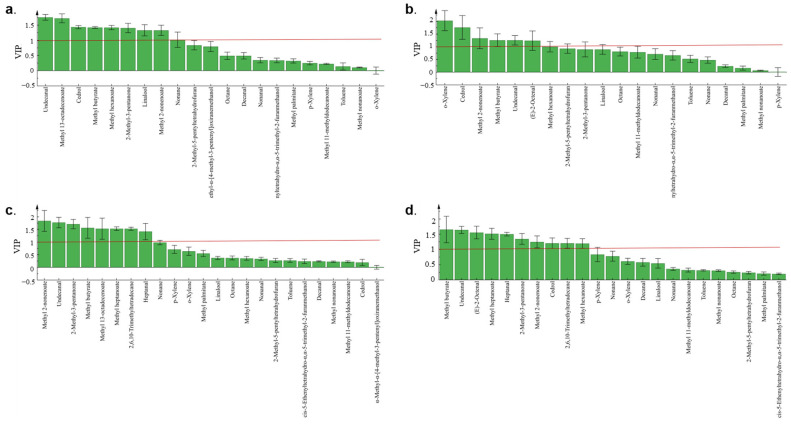
The variable importance in projection (VIP) scores of volatile compounds of mature and immature jujube honey in different contamination stages. (**a**) HMH; (**b**) LMH; (**c**) HIH; (**d**) LIH.

**Figure 5 foods-12-02730-f005:**
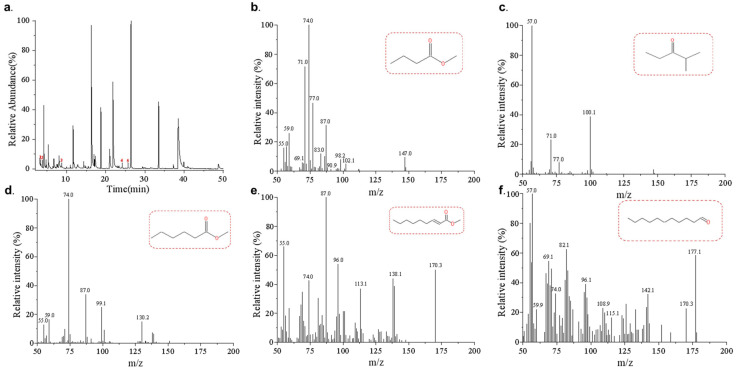
HS-SPME/GC–MS chromatograms of signature VOCs in *Z. rouxii*-contaminated jujube honey. (**a**) Typical TICs of mature jujube honey after *Z. rouxii* contamination: (1) methyl butyrate, (2) 2-Methyl-3-pentanone, (3) methyl hexanoate, (4) methyl 2-nonenoate, (5) undecanal; (**b**) extraction ion flow diagram and chemical structure for signature violate methyl butyrate; (**c**) extraction ion flow diagram and chemical structure for signature violate 2-methyl-3-pentanone; (**d**) extraction ion flow diagram and chemical structure for signature violate methyl hexanoate; (**e**) extraction ion flow diagram and chemical structure for signature violate methyl 2-nonenoate; (**f**) extraction ion flow diagram and chemical structure for signature violate undecanal.

**Figure 6 foods-12-02730-f006:**
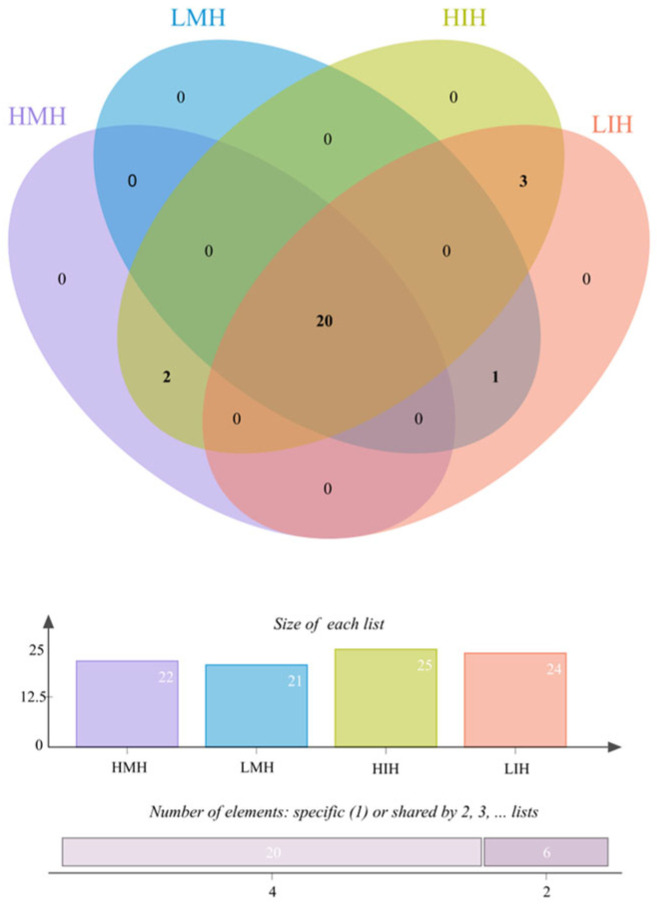
Venn diagram of volatile differences for four groups of *Z. rouxii*-contaminated jujube honey via HS-SPME-GC-MS.

**Table 1 foods-12-02730-t001:** List of VOCs identified via HS-SPME-GC-MS in *Zygosaccharomyces rouxii*-contaminated jujube honey stored for 3–30 days.

	Metabolite	Formula	RT (min)	*m*/*z*	ck ^1^	3d ^1^	7d ^1^	15d ^1^	30d ^1^	ck ^2^	3d ^2^	7d ^2^	15d ^2^	30d ^2^
Alcohols	cis-5-Ethenyltetrahydro-α,α-5-trimethyl-2-furanmethanol	C_10_H_18_O_2_	14.822	170.13	+	+	+	+	+	+	−	+	+	−
	α-Methyl-α-[4-methyl-3-pentenyl]oxiranemethanol	C_10_H_18_O_2_	15.522	170.13	+	+	+	+	+	−	−	−	−	−
	Linalool	C_10_H_18_O	16.143	154.14	−	+	+	+	+	−	−	−	−	+
	Cedrol	C_15_H_26_O	37.259	222.37	−	−	+	+	+	−	−	+	+	−
Aldehydes	(*E*)-2-Octenal	C_8_H_14_O	14.316	126.20	−	−	−	−	−	−	+	+	+	+
	Nonanal	C_9_H_18_O	16.377	142.24	+	+	+	+	+	+	+	+	+	+
	Decanal	C_10_H_20_O	21.024	156.27	+	+	+	+	+	+	+	+	+	+
	Undecanal	C_11_H_22O_	25.660	170.29	−	+	+	+	−	−	+	+	+	−
Ketones	2-Methyl-3-pentanone	C_6_H_12_O	3.834	316.04	−	+	+	+	+	−	−	−	−	+
	Benzophenone *	C_13_H_10_O	38.511	182.22	+	+	+	+	+	+	+	+	+	+
Esters	Methyl butyrate	C_5_H_10_O_2_	3.396	102.13	−	−	+	+	+	−	−	−	−	+
	Methyl hexanoate	C_7_H_14_O_2_	8.675	130.19	−	−	+	+	+	−	−	−	−	+
	Methyl nonanoate	C_10_H_20_O_2_	21.511	172.26	+	+	+	+	+	+	+	+	+	+
	Methyl 2-nonenoate	C_10_H_18_O_2_	21.834	170.25	−	+	+	+	+	−	+	+	+	+
	Methyl 11-methyldodecanoate	C_13_H_26_O_2_	34.535	214.34	+	+	+	+	+	+	+	+	+	+
	Methyl palmitate	C_17_H_34_O_2_	48.825	270.45	+	+	+	+	+	+	+	+	+	+
	Methyl 13-octadecenoate	C_19_H_36_O_2_	54.273	296.49	−	−	−	+	+	−	−	−	−	−
Aromatic	Toluene	C_7_H_8_	4.139	92.14	+	+	+	+	+	+	+	+	+	+
	*p*-Xylene	C_8_H_10_	6.722	106.17	+	+	+	+	+	+	+	+	+	+
	*o*-Xylene	C_8_H_10_	7.465	106.17	+	+	+	+	+	+	+	+	+	+
Alkanes	Octane	C_8_H_18_	4.756	114.23	+	+	+	+	+	+	+	+	+	+
	Nonane	C_9_H_20_	7.677	128.26	+	+	+	+	+	+	+	+	+	+
Furan	2-Methyl-5-pentyltetrahydrofuran	C_10_H_20_O	7.367	156.26	+	+	+	+	+	+	+	+	+	+

Denotes: *: Benzophenone used as internal standard; ^1^: Concentration of 10^−3^
*Z. rouxii* suspension; ^2^: Concentration of 10^−4^
*Z. rouxii* suspension group; “ck”: The blank of no *Z. rouxii* was inoculated; “+”: The substance was detected; “−”: The substance was not detected.

**Table 2 foods-12-02730-t002:** List of VOCs identified by HS-SPME-GC-MS in *Zygosaccharomyces rouxii*-contaminated immature jujube honey stored for 3–30 days.

	Metabolite	Formula	RT (min)	*m*/*z*	ck ^1^	3d ^1^	7d ^1^	15d ^1^	30d ^1^	ck ^2^	3d ^2^	7d ^2^	15d ^2^	30d ^2^
Alcohols	cis-5-Ethenyltetrahydro-α,α-5-trimethyl-2-furanmethanol	C_10_H_18_O_2_	14.822	170.13	+	+	+	+	+	+	+	+	+	+
	α-Methyl-α-[4-methyl-3-pentenyl]oxiranemethanol	C_10_H_18_O_2_	15.522	170.13	+	+	+	+	+	−	−	−	−	−
	Linalool	C_10_H_18_O	16.143	154.14	+	+	+	+	+	+	+	+	+	+
	Cedrol	C_15_H_26_O	37.259	222.37	+	+	+	+	+	+	−	−	−	−
Aldehydes	Heptanal	C_7_H_14_O	7.792	114.19	+	+	+	+	+	−	−	+	+	−
	(*E*)-2-Octenal	C_8_H_14_O	14.316	126.20	−	−	−	−	−	−	+	+	+	+
	Nonanal	C_9_H_18_O	16.377	142.24	+	+	+	+	+	+	+	+	+	+
	Decanal	C_10_H_20_O	21.024	156.27	+	+	+	+	+	+	+	+	+	+
	Undecanal	C_11_H_22O_	25.660	170.29	−	+	+	−	−	−	−	+	+	−
Ketones	2-Methyl-3-pentanone	C_6_H_12_O	3.834	316.04	−	−	+	+	+	−	+	+	+	+
	Benzophenone *	C_13_H_10_O	38.511	182.22	+	+	+	+	+	+	+	+	+	+
Esters	Methyl butyrate	C_5_H_10_O_2_	3.396	102.13	−	−	−	+	+	−	−	−	−	+
	Methyl hexanoate	C_7_H_14_O_2_	8.675	130.19	+	+	+	+	+	−	+	+	+	+
	Methyl heptanoate	C_8_H_16_O_2_	12.838	144.21	+	+	−	−	−	+	+	−	−	−
	Methyl nonanoate	C_10_H_20_O_2_	21.511	172.26	+	+	+	+	+	+	+	+	+	+
	Methyl 2-nonenoate	C_10_H_18_O_2_	21.834	170.25	−	+	+	+	+	−	+	+	+	+
	Methyl 11-methyldodecanoate	C_13_H_26_O_2_	34.535	214.34	+	+	+	+	+	+	+	+	+	+
	Methyl palmitate	C_17_H_34_O_2_	48.825	270.45	+	+	+	+	+	+	+	+	+	+
	Methyl 13-octadecenoate	C_19_H_36_O_2_	54.273	296.49	−	−	−	−	+	−	−	−	−	−
Aromatic	Toluene	C_7_H_8_	4.139	92.14	+	+	+	+	+	+	+	+	+	+
	*p*-Xylene	C_8_H_10_	6.722	106.17	+	+	+	+	+	+	+	+	+	+
	*o*-Xylene	C_8_H_10_	7.465	106.17	+	+	+	+	+	+	+	+	+	+
Alkanes	Octane	C_8_H_18_	4.756	114.23	+	+	+	+	+	+	+	+	+	+
	Nonane	C_9_H_20_	7.677	128.26	+	+	+	+	+	+	+	+	+	+
	2,6,10-Trimethyltetradecane	C_17_H_36_	31.890	240.47	+	+	−	−	−	+	−	−	−	−
Furan	2-Methyl-5-pentyltetrahydrofuran	C_10_H_20_O	7.367	156.26	+	+	+	+	+	+	+	+	+	+

Denotes: *—benzophenone used as internal standard; ^1^—concentration of 10^−3^
*Z. rouxii* suspension; ^2^—concentration of 10^−4^
*Z. rouxii* suspension; “ck”—the blank group of no *Z. rouxii* was inoculated; “+”—the substance was detected; “−”—the substance was not detected.

## Data Availability

Data is contained within the article.

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
