# Peer review of "Identification of Volatile Markers during Early Zygosaccharomyces rouxii Contamination in Mature and Immature Jujube Honey"

_foods, 2023, doi:10.3390/foods12142730_

Round 1

Reviewer 1 Report

Simple corrections are needed, related to the formatting of spaces between words, or words in full, such as "d" or days. Note and correct spaces in IUPAC nomenclature.

Undergo review to ensure the best quality of the article.

Author Response

1.Simple corrections are needed, related to the formatting of spaces between words, or words in full, such as "d" or days. Note and correct spaces in IUPAC nomenclature.

Response: We appreciate this reviewer’s comments. It has been revised based on the comment. And all compound names have been unified based on IUPAC nomenclature.

Reviewer 2 Report

This manuscript investigates the potential of using volatile markers in determination of Z.rouxii early contamination in mature and immature jujube honey. It is an interesting study that plays a crucial role in the food safety aspect. The article is fairly written in general and requires justifications (to improve readability and easy interpretation) to the questions as follows:

1) Introduction: Why focus on only jujube honey and does Z.rouxii found present in other types of honey too? How do volatile organic compounds relate to the contamination? Lack of elaboration. Justify. 

2) As mentioned in line 39, early contamination may not be easy to detect. What are the considerations taken while constructing this classification model (for its sensitivity and accuracy)? 

3) Methodology: For this type of study, a large sample size is commonly required in constructing a reliable model. Please provide the information for sample size (training set and testing set for classification model). And what are the criteria in identifying the authenticity of the genuine samples? Justify.

4) Results & discussion:

line 171, A total of 22, 25?

line 338, for the substances not detected in ripe/ mature honey, will they be excluded in constructing the model as one of the markers? 

Results are meaningful but discussion lack of association between contamination and presence of volatiles, and how they contribute to the accuracy of the detection.

5) Format: Please prepare the manuscript according to the journal's guidelines. References are presented in both authors name and numbering at the same time. Eg line 105, etc

There are minor syntax errors throughout the entire manuscript. Avoid using first person in academic writing too. 

Author Response

  1. Introduction: Why focus on only jujube honey and doesrouxii found present in other types of honey too? How do volatile organic compounds relate to the contamination? Lack of elaboration. Justify. 

Response: We appreciate this reviewer’s comments. Just as the reviewer said, Z.rouxii could widely be detected in all kinds of honey. However, as the VOC of honey varies greatly among different honeys, and due to the limitation of word count and space, this study mainly focuses on the VOC of jujube honey. Related references (volatile organic compounds relate to the contamination) have been added and revised based on the comment. Line 78-80.

  1. As mentioned in line 39, early contamination may not be easy to detect. What are the considerations taken while constructing this classification model (for its sensitivity and accuracy)? 

Response: We thank for the reviewer’s comments. The primary objective of this study was to identify the odor changes and signature volatiles of mature and immature jujube honey during the earliest phases of Z. rouxii contamination based on HS-SPME-GC-MS techniques. Since there is no study at this stage to explore the detection of Z. spp contamination in honey based on VOC, this study was conducted using this method to find possible markers for further rapid detection. And we also found this method could effectively discriminated the Z. rouxii contamination based on PCA and OPLS-DA. However, as the meaningful suggestion, we would like to investigate detail sensitivity of this method based on the findings of this work

  1. Methodology: For this type of study, a large sample size is commonly required in constructing a reliable model. Please provide the information for sample size (training set and testing set for classification model). And what are the criteria in identifying the authenticity of the genuine samples? Justify.

Response: We thank for the reviewer’s comments. Just as we mentioned, the primary objective of this study was to find and identify the odor changes and VOC markers of mature and immature jujube honey during the earliest phases of Z.rouxii contamination based on HS-SPME-GC-MS techniques. Since there is no study at this stage to explore the detection of Z. spp contamination in honey based on VOC, this study was conducted using this method to find possible markers for further rapid detection. And we also found this method could effectively discriminated the Z.rouxii contamination based on PCA and OPLS-DA. However, as the meaningful suggestion, we would like to investigate detail sensitivity of this method based on the findings of this work .

  1. Results & discussion:

line 171, A total of 22, 25?

Response: We thank for the reviewer’s comments. It has been revised to avoid misleading. Line 182

  1. line 338, for the substances not detected in ripe/ mature honey, will they be excluded in constructing the model as one of the markers? 

Response: The VOC markers were selected based on the variable importance in the projection (VIP) quantified the contribution of each component to the classification. And VOCs with VIP values >1 could be identified as potential characteristic markers. We discussed mature and non-mature honeys separately in our VIP analysis. Therefore, markers in mature honey were only for substances contained in mature honey, and substances detected only in immature honey have no effect on it.

  1. Results are meaningful but discussion lack of association between contamination and presence of volatiles, and how they contribute to the accuracy of the detection.

Response: We thank for the reviewer’s comments. Due to the large variety of substances in volatiles, not all of them have been reported with clear correlations. At this stage, this VOC-based detection method for pollutants is relatively new and the related literature is limited, and studies remains to be further explored. On the other hand, at present, we mainly use the approach of existing literature combined with our experimental results to discuss the potential transformation processes between substances. This approach has also been more widely used in the literature related to microbial detection in food based on VOC (some of them listed below). Our primary goal is to find the signature volatiles, and later we will further investigate these signature volatiles in combination with histological analysis, fatty acid analysis, and other related studies to target specific pathways or specific classes of substances.

Related References:

  1. Dixon, B., Ahmed, W. M., Mohamed, A. A., Felton, T., & Fowler, S. J. (2022). Metabolic phenotyping of acquired ampicillin resistance using microbial volatiles from Escherichia coli cultures. Journal of Applied Microbiology. 133(4), 2445-2456. http://doi.org/10.1111/jam.15716.
  2.  Drees, C., Vautz, W., Liedtke, S., Rosin, C., Althoff, K., Lippmann, M., … Kunze-Szikszay, N. (2019). GC-IMS headspace analyses allow early recognition of bacterial growth and rapid pathogen differentiation in standard blood cultures. Applied Microbiology and Biotechnology. 103(21-22), 9091-9101. http://doi.org/ 10.1007/ s00253-019-10181-x.

3. Lu, Y., Zeng, L., Li, M., Yan, B., Gao, D., Zhou, B., … He, Q. (2022). Use of GC-IMS for detection of volatile organic compounds to identify mixed bacterial culture medium. AMB Express. 12(1), Article 31. http://doi.org/ 10.1186/s13568-022-01367-0.

4. Wen, R., Kong, B., Yin, X., Zhang, H., & Chen, Q. (2022). Characterisation of flavour profile of beef jerky inoculated with different autochthonous lactic acid bacteria using electronic nose and gas chromatography-ion mobility spectrometry. Meat Science. 183, Article 108658. http://doi.org/10.1016/j.meatsci.2021.108658.

  1. Format: Please prepare the manuscript according to the journal's guidelines. References are presented in both authors name and numbering at the same time. Eg line 105, etc

Response: We thank for the reviewer’s comments. It has been revised based on the comment. Line 114

Reviewer 3 Report

Abstract: Should include background, objective, materials and methods, results, and conclusion.

Introduction:

·       A paragraph about the chemical properties of jujube honey needs to add.

·       Line 85: please delete (Z. rouxii).

·       Line 126: please delete (2022).

Materials and Methods

·       Line 85: please write the Ref. Of SPSS.

Results and Discussion

·       Line 175: please replace (Table 1, Table 2) with (Table 1and 2).

·       Tables 1 and 2: Please explain + and -.

·       Line 226: please replace Gaglio (2017) with Gaglio [number].

·       Line 257-258: please replace d with days.

·       Line 291:  please replace Zhu et al. (2022) with Zhu et al. [No].

·       Line 446: please replace Martnez-Garca et al. (2021) with Martnez-Garca et al. [No].

·       The authors mentioned "One-way significant differences (P < 0.05) and correlation coefficients were calculated using SPSS 25.0 software" in data analysis. At the same time, I can't find the analysis data and correlation coefficient values on the Tables or Figs. Where them?

·       Fig 4: I noticed error bars are higher than the means in some bars?????

References

· Please write the Ref. Of SPSS.

·       Scientific names should be in italic form.

·       The authors write honeybee(s) as one word and sometimes as two words. Please use one form throughout the manuscript.

Moderate editing of English language required

Author Response

  1. Abstract:Should include background, objective, materials and methods, results, and conclusion.

Response: We appreciate this reviewer’s comments. The abstract has been revised based on the comment. Line 10-23

  1. Introduction:A paragraph about the chemical properties of jujube honey needs to add.

Response: We appreciate this reviewer’s comments. It has been revised based on the comment. Line 62-65

  1.   Line 85: please delete (Z. rouxii).

Response: Done

  1. Line 126: please delete (2022).

Response: Done

  1. Materials and Methods:Line 85: please write the Ref. Of SPSS.

Response: Done. Line 170ï¼›Ref.25

  1. Results and Discussion:Line 175: please replace (Table 1, Table 2) with (Table 1and 2).

Response: Done. Line 198

  1.  Tables 1 and 2: Please explain + and -.

Response: We appreciate this reviewer’s comments. It has been revised based on the comment.

  1. Line 226: please replace Gaglio (2017) with Gaglio [number].

Response: Done. Line 248.

  1. Line 257-258: please replace d with days.

Response: Done. Line 279.

  1.  Line 291:  please replace Zhu et al. (2022) with Zhu et al. [No].

Response: Done. Line 312.

  1.  Line 446: please replace Martnez-Garca et al. (2021) with Martnez-Garca et al. [No].

Response: Done. Line 464

  1. The authors mentioned "One-way significant differences (P < 0.05) and correlation coefficients were calculated using SPSS 25.0 software" in data analysis. At the same time, I can't find the analysis data and correlation coefficient values on the Tables or Figs. Where them?

Response: We appreciate this reviewer’s comments. The detailed information of SD values and correlation coefficient values were listed in supplementary tables. Tables. S1-S4.

  1.  Fig 4: I noticed error bars are higher than the means in some bars?????

Response: We appreciate this reviewer’s comments. The data and figures have been checked and revised.

  1. References· Please write the Ref. Of SPSS.

Response: Done. Ref. 25

  1.  Scientific names should be in italic form.

Response: Done

  1. The authors write honeybee(s) as one word and sometimes as two words. Please use one form throughout the manuscript.

Response: We appreciate this reviewer’s comments. It has been revised based on the comment.

  1. Comments on the Quality of English Language Moderate editing of English language required

Response: We appreciate this reviewer’s comments. The manuscript was thoroughly revised by native speakers and MDPI editing company.

Reviewer 4 Report

Dear Editors and authors, 

Major comments:

1-Why was this isolate selected from yeasts?

2-In work methods, the authors mentioned that yeast was isolated from honey, and they did not indicate whether it is an isolated locality or a strain, and what are the methods used to identify it.

Minor comments:

1-The abstract of the manuscript needs to add some of the results obtained during the study. The final conclusion in the abstract needs some modifications because the compounds that have been identified may be due to other contaminated yeasts.

2-The introduction in the manuscript needs to be supported by some scientific references about the use of the SPME tool in the diagnosis of metabolites resulting from microorganisms, such as

Verma, D. K., Al-Sahlany, S. T. G., Niamah, A. K., Thakur, M., Shah, N., Singh, S., ... & Aguilar, C. N. (2022). Recent trends in microbial flavour Compounds: A review on Chemistry, synthesis mechanism and their application in food. Saudi Journal of Biological Sciences, 29(3), 1565-1576.‏

Carraturo, F., Libralato, G., Esposito, R., Galdiero, E., Aliberti, F., Amoresano, A., ... & Guida, M. (2020). Metabolomic profiling of food matrices: Preliminary identification of potential markers of microbial contamination. Journal of food science, 85(10), 3467-3477.‏

Mhlongo, M. I., Piater, L. A., & Dubery, I. A. (2022). Profiling of volatile organic compounds from four Plant Growth-Promoting Rhizobacteria by SPME–GC–MS: A metabolomics study. Metabolites, 12(8), 763.‏

3-The objective of the manuscript is unclear. A clear objective should be written for the work of the manuscript.

4-How was it confirmed that the mature and immature honey used in the experiment was not originally contaminated with this yeast or another yeast?

The authors had to do a microbial examination of the honey before starting the experiment.

5-Table No. 1 and Table No. 2 Each table must contain an explanation of the symbols in it written at the bottom of the table, such as the meaning of ck.

6-The authors did not indicate in Tables 1 and 2 which compound is the most abundant or most prevalent among the compounds .The sign (+) means the compound is present, but what is its amount or quantity among other compounds. Are all compounds the same?

7-Figure 2 and 3 The x- and y-axis are not clear as well as the units of measurement.

8-Figure 5 The y-axis is not clear, but the x-axis is unlabeled.

9-Conclusions contain many results. Conclusions must be rewritten again.

The language of the manuscript is simple, good and easily readable

Author Response

  1. Why was this isolate selected from yeasts?

Response: We appreciate this reviewer’s comments. In our previous study, a total of seven osmotolerant yeast strains were isolated from 35 honey samples collected from three provinces between 2013 and 2017. All of them were identified as Zygosaccharomyces spp. by 26S rDNA. Among them, the most prevalent specie was Zygosaccharomyces rouxii. This section is not expanded in detail here because it is covered in another work. In consideration of similar studies (Chen et al., 2022), this strain was finally selected for the study.

Reference: Chen, SQ., Tang, QY., Geng, JQ., Liu, YQ., Jiang, J., Cai, XF., Cao, H., Wu, YT., Ren, Y., Liu K. 2022. Detection of Viable Zygosaccharomyces rouxii in Honey and Honey Products via PMAXX-qPCR. Journal of food Quality. Article:8670182. https://doi.org/10.1155/2022/8670182

  1. In work methods, the authors mentioned that yeast was isolated from honey, and they did not indicate whether it is an isolated locality or a strain, and what are the methods used to identify it.

Response: We appreciate this reviewer’s comments. The Zygosaccharomyces rouxii used in the experiment was screened in the previous study, and was identified as Zygosaccharomyces rouxii by 26SrDNA after isolation and purification. Since this part is described in detail in another paper, it was not mentioned in this paper to prevent repetition of the expression. We have added some detail information in the methods based on the comments. Line 114-117

Minor comments:

  1. The abstract of the manuscript needs to add some of the results obtained during the study. The final conclusion in the abstract needs some modifications because the compounds that have been identified may be due to other contaminated yeasts.

Response: We appreciate this reviewer’s comments. Detail results has been added in abstract based on the comment.

Since the honey samples used in this study were tested and isolated for yeasts using the plate counting method with YPD medium based on Guo et al. (2018). It was ensured that the honey samples were not contaminated with Zygosaccharomyces rouxii. before the experiment. Besides, honey samples without inoculation to yeast (CK) were used in the study to be tested after 30 days under the same conditions as a blank control to ensure that the results of this study were not contaminated or interfered by other yeasts. Therefore, it is highly unlikely that this change was caused by other yeasts. However, in case of misleading, this part has been added to the methods based on the comments. Line 18-21; Line 114-117.

  1. The introduction in the manuscript needs to be supported by some scientific references about the use of the SPME tool in the diagnosis of metabolites resulting from microorganisms, such as
  2. Verma, D. K., Al-Sahlany, S. T. G., Niamah, A. K., Thakur, M., Shah, N., Singh, S., ... & Aguilar, C. N. (2022). Recent trends in microbial flavour Compounds: A review on Chemistry, synthesis mechanism and their application in food. Saudi Journal of Biological Sciences, 29(3), 1565-1576. 21. Carraturo, F., Libralato, G., Esposito, R., Galdiero, E., Aliberti, F., Amoresano, A., ... & Guida, M. (2020). Metabolomic profiling of food matrices: Preliminary identification of potential markers of microbial contamination. Journal of food science, 85(10), 3467-3477. .‏22. Mhlongo, M. I., Piater, L. A., & Dubery, I. A. (2022). Profiling of volatile organic compounds from four Plant Growth-Promoting Rhizobacteria by SPME–GC–MS: A metabolomics study. Metabolites, 12(8), 763.‏

Response: We appreciate this reviewer’s comments. References has been cited based on the comment. Ref. 20, 21, 22.

  1. The objective of the manuscript is unclear. A clear objective should be written for the work of the manuscript.

Response: We appreciate this reviewer’s comments. It has been revised in introduction based on the comment. Line 95-98.

  1. How was it confirmed that the mature and immature honey used in the experiment was not originally contaminated with this yeast or another yeast?The authors had to do a microbial examination of the honey before starting the experiment.

Response: We appreciate this reviewer’s comments. The honey samples used in this study were tested and isolated for yeasts using the plate counting method with YPD medium based on Guo et al. (2018). It was ensured that the honey samples were not contaminated with Zygosaccharomyces rouxii. before the experiment. Besides, honey samples without inoculation to yeast (CK) were used in the study to be tested after 30 days under the same conditions as a blank control to ensure that the results of this study were not contaminated or interfered by other yeasts. We have added this part of the expression to the methods based on the comments. Line 114-117.

Reference: Guo, H., Yuan, YH., Niu, C., Qiu, Y., Wei, JP., Yue, TL. 2018. Development of an indirect enzyme-linked immunosorbent assay for the detection of osmotolerant yeast Zygosaccharomyces rouxii in different food

  1. Table No. 1 and Table No. 2 Each table must contain an explanation of the symbols in it written at the bottom of the table, such as the meaning of ck.

Response: We appreciate this reviewer’s comments. It has been revised based on the comment.

8. The authors did not indicate in Tables 1 and 2 which compound is the most abundant or most prevalent among the compounds .The sign (+) means the compound is present, but what is its amount or quantity among other compounds. Are all compounds the same?

Response: We appreciate this reviewer’s comments. We performed a semi-quantitative assay of the compounds, where the specific data of the changes in the relative content of the substances were listed in Supplementary tables and Figure1. (Tables S1, S2, S3, and S4). Besides, the block colors in fig.1 also reflect the changes in the relative content of the substances.

9.Figure 2 and 3 The x- and y-axis are not clear as well as the units of measurement.

Response: We appreciate this reviewer’s comments. Figures have been revised based on the comment.

  1. Figure 5 The y-axis is not clear, but the x-axis is unlabeled.

Response: We appreciate this reviewer’s comments. Figures have been revised based on the comment.

Round 2

Reviewer 3 Report

Thanks for improving the manuscript.

Minor editing of English language required.

Reviewer 4 Report

Dear Editors, 

The authors have made all required corrections and modifications to the manuscript. I think the manuscript is ready for publication.

The language of the manuscript is good and easy to read.